# Predicting future community-level ocular *Chlamydia trachomatis* infection prevalence using serological, clinical, molecular, and geospatial data

**Christine Tedijanto** [1] *, **Solomon Aragie** [2], **Zerihun Tadesse** [2], **Mahteme Haile** [3], **Taye Zeru** [3], **Scott D. Nash** [4], **Dionna M. Wittberg** [1], **Sarah Gwyn** [5], **Diana L. Martin** [5], **Hugh J. W. Sturrock** [6], **Thomas M. Lietman** [1,7,8,9], **Jeremy D. Keenan** [1,7], **Benjamin F. Arnold** [1,7]

1 Francis I. Proctor Foundation, University of California, San Francisco, California, United States of America, 2 The Carter Center Ethiopia, Addis Ababa, Ethiopia, 3 Amhara Public Health Institute, Bahir Dar, Ethiopia, 4 The Carter Center, Atlanta, Georgia, United States of America, 5 Division of Parasitic Diseases and Malaria, Centers for Disease Control and Prevention, Atlanta, Georgia, United States of America, 6 Locational, Poole, United Kingdom, 7 Department of Ophthalmology, University of California, San Francisco, California, United States of America, 8 Department of Epidemiology and Biostatistics, University of California, San Francisco, California, United States of America, 9 Institute for Global Health Sciences, University of California, San Francisco, California, United States of America

* christine.tedijanto@ucsf.edu

**Data Availability Statement:** Community latitude and longitude values have been modified to protect the privacy of study participants. The pre-specified

## Abstract

Trachoma is an infectious disease characterized by repeated exposures to *Chlamydia trachomatis* (*Ct*) that may ultimately lead to blindness. Efficient identification of communities with high infection burden could help target more intensive control efforts. We hypothesized that IgG seroprevalence in combination with geospatial layers, machine learning, and model-based geostatistics would be able to accurately predict future community-level ocular *Ct* infections detected by PCR. We used measurements from 40 communities in the hyperendemic Amhara region of Ethiopia to assess this hypothesis. Median *Ct* infection prevalence among children 0–5 years old increased from 6% at enrollment, in the context of recent mass drug administration (MDA), to 29% by month 36, following three years without MDA. At baseline, correlation between seroprevalence and *Ct* infection was stronger among children 0–5 years old (ρ = 0.77) than children 6–9 years old (ρ = 0.48), and stronger than the correlation between active trachoma and *Ct* infection (0-5y ρ = 0.56; 6-9y ρ = 0.40). Seroprevalence was the strongest concurrent predictor of infection prevalence at month 36 among children 0–5 years old (cross-validated $R^2$ = 0.75, 95% CI: 0.58–0.85), though predictive performance declined substantially with increasing temporal lag between predictor and outcome measurements. Geospatial variables, a spatial Gaussian process, and stacked ensemble machine learning did not meaningfully improve predictions. Serological markers among children 0–5 years old may be an objective tool for identifying communities with high levels of ocular *Ct* infections, but accurate, future prediction in the context of changing transmission remains an open challenge.

statistical analysis plan is available on Open Science Framework (https://osf.io/t48zb/). De-identified data and code to replicate this work are available at the following repository: https://doi.org/10.5281/zenodo.5851642.

**Funding:** This work was supported by the National Institute of Allergy and Infectious Diseases (R03 AI147128 to BFA) and the National Eye Institute (U10 EY023939 to JDK). This work was also made possible in part by an Unrestricted Grant from Research to Prevent Blindness. The funders had no role in study design, data collection and analysis, decision to publish, or preparation of the manuscript.

**Competing interests:** The authors have declared that no competing interests exist.

## Author summary

Trachoma, one of the leading infectious causes of blindness globally, is targeted for elimination as a public health problem by 2030. District-level estimates of active trachoma among children 1–9 years old are currently used to guide control programs and assess elimination. However, active trachoma, based on diagnosis of clinical signs, is a subjective indicator. Serological markers present an objective, scalable alternative that could be measured in integrated platforms. In a hyperendemic region, community-level seroprevalence aligned more closely with concurrent infection prevalence than active trachoma. The correlation between seroprevalence and infection prevalence was stronger among 0–5-year-olds compared to 6–9-year-olds and was consistent over a three-year period of increasing transmission. Serosurveillance among children 0–5 years old may be a promising monitoring strategy to identify communities with the highest burdens of ocular chlamydial infection.

## Introduction

Trachoma, caused by ocular infection with the bacterium *Chlamydia trachomatis* (*Ct*), is a leading infectious cause of blindness worldwide [1] and has been targeted for elimination as a public health problem by 2030 [2]. The World Health Organization's SAFE strategy (Surgery, Antibiotics, Facial cleanliness, and Environmental improvement) has been successful in countries across Asia and the Middle East, achieving elimination as a public health problem in many cases [2]. Yet, trachoma is a persistent challenge in pockets of Africa, including some areas of Ethiopia that remain hyperendemic despite over 10 years of control activities [3]. The ability to efficiently identify potential areas of ongoing transmission for follow-up surveys and more intensive interventions is crucial for the trachoma endgame.

Trachoma elimination programs are currently guided by estimates of active trachoma in evaluation units (EUs) of 100,000–250,000 people [4]. Evidence of trachoma clusters at the village- or sub-village level throughout Africa [5–10] suggest that aggregate estimates may mask heterogeneity in infection: high-transmission villages may be missed by sampling design or their signal may be "washed out" in EU-level averages. Fine-scale estimates of trachoma could facilitate targeted allocation of limited resources to communities with the highest burden [11] and reduce unnecessary antibiotic use and subsequent selection for antibiotic resistance [12].

Mass drug administration (MDA) of azithromycin is currently recommended for EUs with trachomatous inflammation—follicular (TF) prevalence ≥5% among children 1–9 years old [2]. Clinical disease states are relevant signals of progression towards conjunctival scarring and ultimately blindness [1] but are subject to misclassification, even by experienced graders [13]. Immunoglobulin G (IgG) antibody responses to Pgp3 and CT694 antigens are a more objective alternative and have been identified as sensitive, specific, and durable indicators of past ocular *Ct* infection [14,15]. In addition, dried blood spot specimens used to assess serological markers are easy to collect, and *Ct* antigens can be included in multiplexed, integrated serosurveillance platforms to simultaneously and cost-effectively monitor numerous pathogens [16].

Thus far, efforts to predict future trachoma prevalence at the village and district level have had modest success [17,18] but have not considered serology or recent advances in machine learning and geostatistics that may facilitate fine-scale prediction. We hypothesized that models incorporating trachoma indicators (active trachoma, ocular *Ct* infection identified by polymerase chain reaction (PCR), and IgG response to *Ct* antigens), remotely sensed geospatial

layers, and spatial structure would accurately predict future community-level *Ct* infection prevalence. We also hypothesized that seroprevalence would be a more accurate and stable predictor of *Ct* infections compared to active trachoma and that communities with high levels of infection would be geographically clustered in stable foci of transmission ("hotspots"). We tested our hypotheses using measurements from 40 communities in the hyperendemic Amhara region of Ethiopia.

## Methods

### Ethics statement

This research was approved by a human subjects review board at the University of California, San Francisco. Each participant or guardian provided verbal consent before any study activity, with separate consent required for census, examinations and intervention at each study visit.

### Data collection

This work was a secondary analysis of data from the WASH Upgrades for Health in Amhara (WUHA) community-randomized trial, one of the trials in the Sanitation, Water, and Instruction in Face-Washing for Trachoma (SWIFT) (NCT02754583) series. Details of study methodology and implementation are described in the published protocol [19]. WUHA was conducted from November 2015 through March 2019 in the Gazgibella, Sekota Zuria (i.e. Sekota) and Sekota Ketema (i.e. Sekota town) woredas of the Wag Hemra Zone in Amhara, Ethiopia (Fig 1). Forty communities were randomized in a 1:1 ratio to receive a comprehensive Water, Sanitation, and Hygiene (WASH) package at baseline or at completion of the study. Communities were not selected at random; they were located in rural areas within a 4-hour drive and/or walk from the main road and included all households within 1.5 km of a potential water point (e.g. hand-dug well or protected spring) as determined by geohydrologic survey; further details are available in the study protocol [19]. Mass administration of azithromycin occurred for seven consecutive years (May 2009 to June 2015, with supplemental administration in October 2014) prior to the start of the study but was suspended in all study communities for the duration of the WUHA trial.

Trachoma indicators were measured in each study community at baseline and three annual monitoring visits. Approximately one month prior to each monitoring visit, a census was taken to enumerate individuals living in each study community. At each visit, thirty individuals in each of three age groups (0–5 years, 6–9 years, 10+ years) were randomly selected from each community for monitoring; this analysis focused on children 0–9 years old. Per the trial design, not all trachoma indicators were measured in all age groups at each time point; only children 0–5 years old were tested for clinical, serological, and PCR outcomes at all visits. At the end of WUHA, after adjusting for baseline, there was no statistically significant difference in the primary endpoint of community-level ocular *Ct* infection among 0–5-year-olds between intervention arms across the three post-baseline time points (risk difference: 3.7 percentage points higher in WASH arm, 95% CI: -4.9 to 12.4, p = 0.40) [20]. As a result, we combined information across arms for this analysis.

### Measurement and definition of trachoma indicators

We analyzed age-group-specific community-level prevalence of three trachoma indicators: active trachoma, ocular Ct infection detected by PCR, and IgG response to Pgp3 and CT694 antigens.

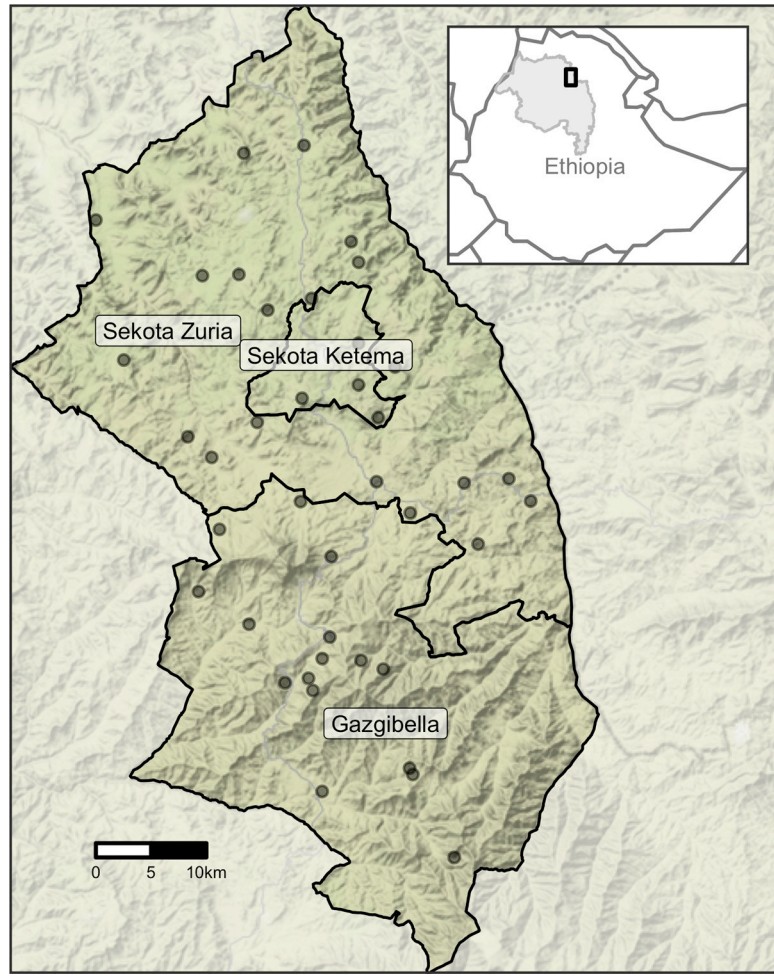

**Fig 1. Map of study area.** Inset (top right) highlights the Amhara Region (gray shading) of Ethiopia and the study area (black rectangle). Forty communities from three woredas (administrative level 3) in Amhara were included in the WUHA trial. The base map layer for this figure was downloaded from Stamen Maps ("Terrain") and is available under the CC BY 3.0 license.

Each year, eight local nurses and other healthcare professionals were recruited to serve as trachoma graders and swabbers. These individuals completed a four-day training with two days of classroom training and two field practice days. Prior to participation in fieldwork, graders were required to pass a photographic grading test with a Cohen's kappa score of 0.6 or greater relative to consensus grades from a panel of three expert graders. Grading teams were randomly assigned to clusters. Trained trachoma graders used a pair of 2.5× loupes and a flashlight to assess the everted right superior tarsal conjunctiva for the presence of trachomatous inflammation—follicular (TF) or trachomatous inflammation—intense (TI) according to the WHO grading system [21]. Specifically, TF is characterized by the presence of five or more follicles which are (a) each at least 0.5 mm in diameter and (b) located in the central part of the upper tarsal conjunctiva. TI is distinguished by pronounced inflammatory thickening of the upper tarsal conjunctiva that obscures more than half of the normal deep tarsal vessels. An individual was considered positive for active trachoma if either TF or TI was detected.

Conjunctival swabs were collected and tested in the study laboratory at the Amhara Public Health Institute in Bahir Dar, Ethiopia with the Abbott RealTime assay (automated Abbott

m2000 System), which is highly sensitive and specific for *Ct* [22,23]. Groups of five samples, stratified by community and age group, were pooled for testing, and community-level *Ct* infection prevalence was estimated from pooled results using a maximum likelihood approach [24]. Swabs from positive pools were tested individually for 0–5-year-olds at all visits, for 6–9-year-olds at months 12, 24, and 36, and if >80% of pools for a cluster were positive for all other age groups and time points. Approximately 12% of samples from 6–9-year-olds with an equivocal or positive pooled result at baseline were also tested individually. Air swabs were collected in every cluster at the beginning and end of each monitoring visit. None of the air swabs tested positive for *Ct*.

To measure antibody response, field staff lanced the index finger of each individual and collected blood onto TropBio filter paper. Samples were tested at the US Centers for Disease Control on a multiplex bead assay on the Luminex platform for antibodies to two recombinant antigens (Pgp3, CT694) that measure previous exposure to *C. trachomatis* [14,15,25]. Seropositivity thresholds were defined as median fluorescence intensity minus background (MFI-bg) of 1113 for Pgp3 and 337 for CT694 using an ROC cutoff from reference samples [26]. Individuals who were seropositive with respect to both antigens were considered seropositive for the main analysis.

### Descriptive analysis of trachoma indicators

Spearman rank correlation coefficients were calculated for pairwise combinations of trachoma indicators by age group and follow-up visit. Correlations were also calculated between PCR prevalence at month 36 and serological, PCR, and active trachoma prevalence at each preceding time point to observe changes in correlation with increasing temporal lag between measurements. 95% confidence intervals were estimated from 1000 bootstrap samples. As communities were the unit of analysis, each bootstrap replicate consisted of forty communities resampled with replacement. This aligns with measures of uncertainty for cluster-level summaries which treat clusters as the primary source of variation [27,28].

### Descriptive spatial analysis

Administrative boundaries for Ethiopia were downloaded from the Humanitarian Data Exchange [29]. Spatially interpolated maps for each trachoma indicator at each time point were generated using a simple kriging model including latitude, longitude, and a Matérn covariance. We estimated empirical variograms after removing linear spatial trends for distances up to 33.3 km (half of the maximum distance between any two study communities) and fit exponential and Matérn models; for stability, we required bins to contain ten or more pairs of communities. The effective, or practical, range was defined as the distance at which the fitted model reached 95% of the sill. We compared the observed variograms to a 95% pointwise envelope based on 1000 Monte Carlo simulations; for each simulation, prevalence residuals were permuted while holding coordinates fixed and the empirical variogram was recalculated [30]. We also calculated Moran's I, a measure of global spatial autocorrelation, over 1000 permutations of the community-level prevalence values and estimated a p-value based on permutations resulting in a Moran's I greater than or equal to the observed value.

### Predictive model selection

Prediction models were limited to children 0–5 years old due to availability of all trachoma indicators for this age range at all time points. We developed several candidate models using baseline data only, with the analysis team masked to any future measurements. A wide range of publicly available environmental [31–35], demographic [36], and socioeconomic [37–39]

variables were explored based on prior associations with trachoma or other infectious diseases (**S1 Table**). When possible, features were extracted and aggregated using Google Earth Engine [40], and means were used for spatial and temporal aggregation unless otherwise specified in **S1 Table**. All features were aggregated to a grid resolution of 2.5 arc minutes (approximately 4.5 km at the median latitude of the study area) based on the lowest resolution dataset (Terra-Climate) and reprojected to WGS84. Each community was assigned to the grid cell containing its household-weighted geographic centroid, defined as the median latitude and longitude across all households in the community.

Models were built using predictor variables measured over the same ("concurrent") and prior ("forward predictions") time periods. Time-varying features were summarized based on calendar year, with 2015 data considered "concurrent" with month 0 trachoma indicators and so on. Time-varying features were first aggregated by month and then summarized based on recency relative to the time of monitoring (e.g. last 1 month or December of the calendar year, last 2 months, up to 12 months). To reduce collinearity, we evaluated pairwise Pearson correlation coefficients between temporal summaries of the same variable and dropped the summary over fewer months for pairs with correlation over 0.9.

During preliminary model development with baseline data, we observed that including a large number of predictor variables led to overfitting and unstable model performance due to the relatively small number of communities. As a result, logistic LASSO regression was used to identify a restricted set of geospatial features to include in the final prediction models. Night light radiance and daily precipitation averaged over the preceding 12 months were selected from a model using concurrently measured predictors and outcomes across all follow-up visits.

Logistic regression models of the following form were used as base prediction models:

$$logit(\pi_{cm}) = \alpha + \sum_p \beta_{np} x_{cnp} + S(latitude_c, longitude_c)$$

where $\pi_{cm}$ represents PCR prevalence for study community $c$ at month $m$, $\alpha$ is the model intercept, and $x_{cn1}...x_{cnp}$ denote $p$ covariates (and corresponding coefficients $\beta$) measured at time $n$, where $n = m$ for concurrent predictions and $n = m—k$ for predictions $k$ months forward. Extended models also included a Gaussian process with Matérn covariance function [41] to capture residual spatial structure, represented by the $S$ function dependent on latitude and longitude of each community.

We additionally explored stacked ensemble machine learning, also known as stacked regression [42] or stacked generalization [43]. Stacked ensembles combine predictions from multiple 'Level 0' models using a 'Level 1' model, also called the superlearner or metalearner [44]. Ensembles are theoretically guaranteed to perform as well as or better than any single member of their library [42,44]. Our 'Level 0' learners included logistic regression, generalized additive models [45], random forest [46], extreme gradient boosting [47], and multivariate adaptive regression splines [48]. This set of models, including parametric, semi-parametric, and tree-based methods, was selected to ensure diversity in approach; outcome specification also varied (e.g. binomial, quasibinomial, continuous) based on requirements of the learner. Logistic regression with a Matérn covariance was used as the 'Level 1' superlearner for the baseline analysis.

## Predictive model assessment

We conducted 10-fold cross-validation to assess predictive performance. Spatial autocorrelation can violate the independence assumption between training and validation sets in cross-validation and lead to overly optimistic estimates of predictive power [49,50]. Therefore, we

partitioned the study area into 12 15x15km blocks, each containing 1–8 spatially proximate communities. Communities in the same block were assigned to the same validation set, with some sets consisting of more than one block. This approach decreases spatial dependence between training and validation sets in the same fold and simulates prediction in a new, but geographically proximate, area. Predictive performance was assessed using cross-validated root-mean-square-error (RMSE) and $R^2$ [51], where $R^2$ was calculated as:

$$1 - \frac{\sum_c (p_{cm} - \widehat{p_{cm}})^2}{\sum_c (p_{cm} - \overline{p_{cm}})^2}$$

95% confidence intervals for $R^2$ were estimated using the influence function [52,53]. Communities received equal weight in all validation metrics.

As this was a secondary analysis, the sample size was fixed at 40 communities per survey. To our knowledge, there are no methods available to estimate power for cross-validated error in prediction problems. Instead, we estimated the minimum detectable effect for the correlation analysis. Assuming a two-tailed alpha of 0.05, we had 80% power to detect a correlation of 0.43 or larger with 40 communities [54].

## Results

### Study population

Approximately thirty children from each of two age groups (0–5 years old and 6–9 years old) were randomly sampled from each community at baseline and follow-up visits. The number of children evaluated differed slightly for each trachoma indicator (S2 Table). Over the three-year study period, ocular Ct infection prevalence, as measured by PCR, increased substantially in both age groups (Table 1). Levels of active trachoma fluctuated with time but remained fairly consistent with baseline levels. Seropositivity, defined as antibody response above pre-determined cut-offs for both Pgp3 and CT694 antigens, increased gradually among 0–5-year-olds (two-sided p = $2.6 \times 10^{-4}$ in a Wilcoxon signed-rank test comparing month 0 and month 36). Antibodies were not measured among 6–9-year-olds at months 12 and 24 but were similar between study arms at months 0 and 36 (p = 0.44). Results were similar when seroprevalence was assessed for each antigen separately (S3 Table).

Ocular infection was more common in the western and northern regions of the study area (**Fig 2A**), and seroprevalence and active trachoma were similarly distributed in space (**S1A and S2A** Figs). Based on empirical variograms (**Fig 2B**) and Moran's I (**Fig 2C**), there was

**Table 1. Community-level prevalence of trachoma across 40 study communities by indicator, age group and month of follow-up visit.**

| Month | Median prevalence (%) (IQR), 0–5-year-olds | | | | Median prevalence (%) (IQR), 6–9-year-olds | | | |
|---|---|---|---|---|---|---|---|---|
| | n[1] | PCR[2] | TF/TI[3] | Serology | n[1] | PCR[2] | TF/TI[3] | Serology[4] |
| 0 | 1,269 | 5.6 (2.9–18.1) | 62.9 (51.0–72.5) | 25.0 (10.1–34.8) | 1,135 | 3.5 (0.0–13.9) | 40.3 (25.9–54.9) | 49.2 (29.8–60.2) |
| 12 | 1,162 | 19.1 (6.6–30.2) | 50.8 (40.6–61.1) | 29.7 (15.6–40.2) | 1,092 | 10.9 (5.7–17.4) | 21.3 (14.3–27.8) | - |
| 24 | 1,214 | 27.4 (11.6–34.3) | 67.5 (55.5–77.4) | 33.3 (20.5–39.0) | 1,208 | 19.9 (9.7–34.2) | 45.1 (29.4–53.4) | - |
| 36 | 1,192 | 29.3 (16.2–46.8) | 56.7 (45.2–64.3) | 33.3 (23.5–42.3) | 1,218 | 21.7 (15.2–38.2) | 38.2 (30.1–53.6) | 50.8 (28.9–65.4) |

1 Number of children tested for any indicator across all study communities

2 Polymerase chain reaction

3 Trachomatous inflammation—follicular / trachomatous inflammation—intense

4 Serology was not measured for a random sample of 6–9-year-olds at months 12 and 24

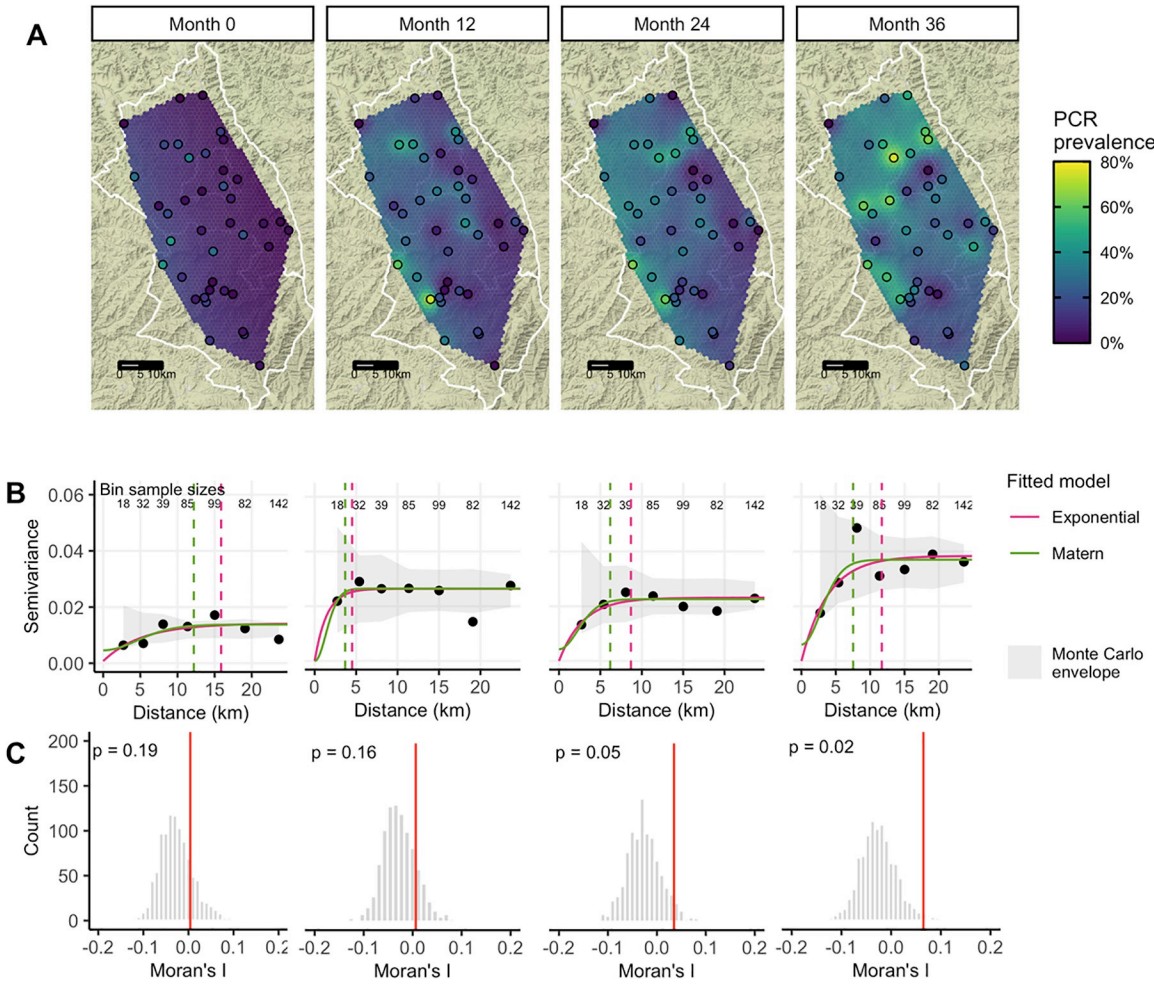

**Fig 2. Predicted surface (A), variograms (B), and Moran's I (C) for PCR-confirmed ocular *C. trachomatis* infection prevalence among 0–5-year-olds at each study month.** Maps display prevalence for 40 study communities at each follow-up visit, spatially interpolated over the convex hull using kriging. Variograms capture similarity between community-level prevalence measurements as a function of distance between community pairs (in km), with smaller semivariance values representing increased similarity. Exponential (magenta) and Matérn (green) models were fit to each empirical variogram, and the effective range (dashed vertical line) is defined as the distance at which the fitted model reaches 95% of the sill. The Monte Carlo envelope (gray shading) displays pointwise 95% coverage of 1000 permutations, representing a null distribution. Moran's I was calculated over 1000 permutations (gray bars, with observed value represented by red line), and a permutation-based p-value was calculated. The base map layer for panel A in this figure was downloaded from Stamen Maps ("Terrain") and is available under the CC BY 3.0 license.

weak spatial structure in community-level *Ct* PCR prevalence that increased slightly over the study period; serology and active trachoma also did not display clear spatial structure over the study area (**S1** and **S2** **Figs**).

## Comparisons between serological, clinical, and molecular trachoma indicators

Seroprevalence demonstrated a stronger rank-preserving relationship, as measured by the Spearman correlation, with contemporaneous PCR prevalence than active trachoma for both age groups (Fig 3A and 3B). Descriptive results were similar when considering either antigen separately (S3 Fig and S3 Table). At baseline, immediately following seven years of MDA, the

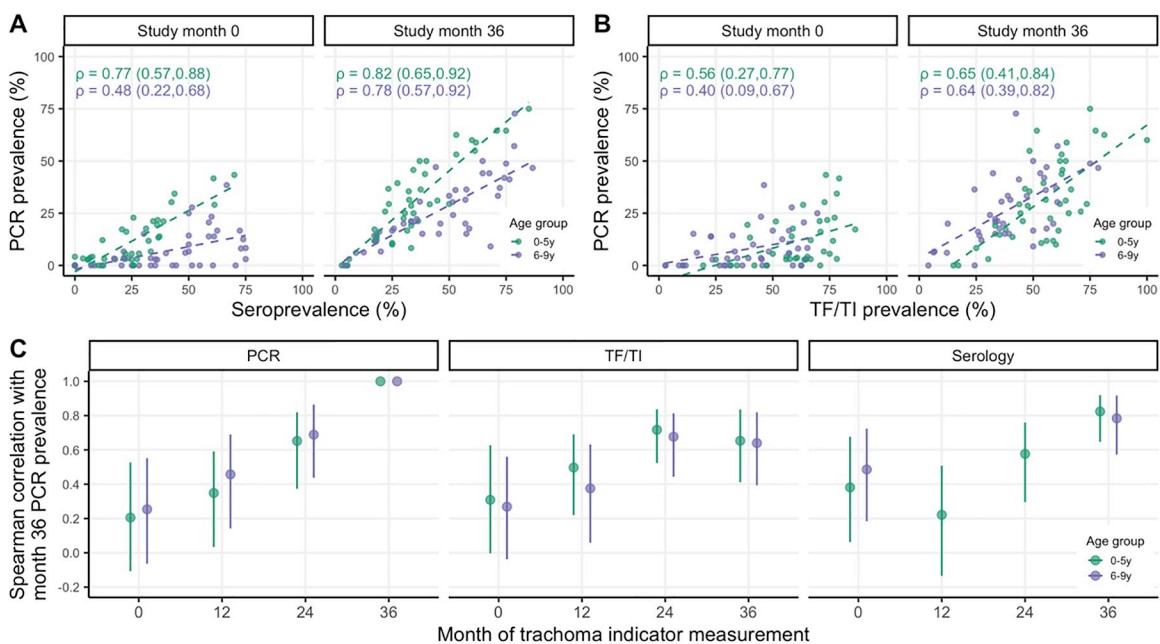

**Fig 3. Correlations between trachoma indicators by age group and over time.** Panels display Spearman rank correlations between community-level seroprevalence and PCR prevalence at study months 0 and 36 (A), active trachoma prevalence and PCR prevalence at months 0 and 36 (B), and PCR prevalence at month 36 and trachoma indicators measured at each survey across 40 study communities (C). Correlations are shown separately for 0–5-year-olds (green) and 6–9-year-olds (purple), and 95% confidence intervals were estimated from 1000 bootstrap samples. Serology data were not collected for a random sample of 6–9-year-olds at months 12 and 24.

correlations between trachoma indicators were more pronounced among younger children, potentially reflecting lower transmission in the presence of MDA and saturation in seroprevalence due to durable antibody responses among older children. Similar saturation dynamics may be at play for active trachoma, which has been shown to resolve slowly among children [55]. By month 36, when infections were higher across the study area (Table 1), correlations between trachoma indicators were similar across age groups (Fig 3A and 3B). Rank-preserving relationships between indicators at each time point and month 36 PCR prevalence were stronger for more proximate measurements, and this increase was more pronounced for PCR compared to active trachoma or serology (Fig 3C).

## Concurrent and forward prediction of PCR prevalence

We predicted community-level infection prevalence using a range of model specifications and conducted spatial 10-fold cross-validation (CV) with 15x15 km blocks [49] to assess predictive performance using CV $R^2$ and root-mean-square-error (RMSE). **Fig 4** presents results for models predicting PCR prevalence at month 36. "Concurrent" predictions utilized trachoma indicators measured at month 36 and/or geospatial variables measured over the preceding year (2018), while "forward" predictions used covariates measured 12, 24, or 36 months in the past. Seroprevalence was the single strongest concurrent predictor of month 36 community-level PCR prevalence (CV $R^2$: 0.75, 95% confidence interval (CI): 0.58–0.85, CV RMSE: 0.10), substantially outperforming active trachoma prevalence (CV $R^2$: 0.37, 95% CI: 0.08–0.56, CV RMSE: 0.16) (**Fig 4**). When predicting 12 months into the future, all trachoma indicators performed moderately well, but predictive performance declined for longer time horizons across all model specifications. No model that we assessed had a CV $R^2$ significantly different from 0

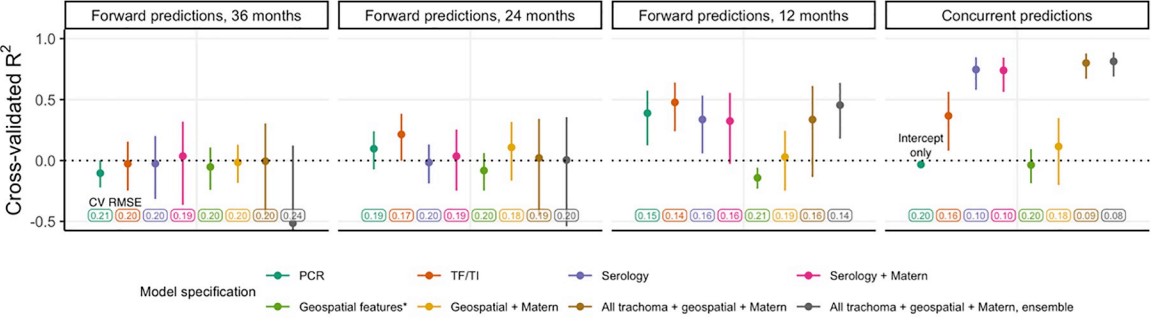

**Fig 4. Cross-validated R² for models predicting month 36 community-level PCR prevalence among 0–5-year-olds.** Cross-validated coefficient of determination ($R^2$), 95% influence-function-based confidence interval, and cross-validated root-mean-square error (RMSE, text label) are shown for each model specification. Logistic regression was used for all models with the exception of the stacked ensemble (gray). Blocks of size 15x15 km were used for 10-fold spatial cross-validation.

(equivalent to an intercept-only or mean-only model) when predicting PCR prevalence 24 months or more into the future.

As anticipated by the weak spatial dependence in PCR prevalence (**Fig 2**), incorporation of a Gaussian process with a Matérn covariance function did not improve predictions. In addition, LASSO-selected geospatial features (night light radiance and daily precipitation averaged over the preceding 12 months) (**S4 Fig**) and a stacked ensemble approach leveraging five base models did not meaningfully improve CV $R^2$ or CV RMSE compared to simpler models. Results were similar for models predicting PCR prevalence at each time point and pooled over all time points (**S5 Fig**). We also observed similar results with various superlearner models (**S6 Fig**) and cross-validation folds (**S7 Fig**), with the latter perhaps reflecting the weak spatial autocorrelation observed in this dataset (**Fig 2**).

## Efficient identification of high-burden communities

A complementary task to prediction is identifying communities with the highest infection burden, defined here as the number of Ct infections among 0–5-year-olds at a given time. To

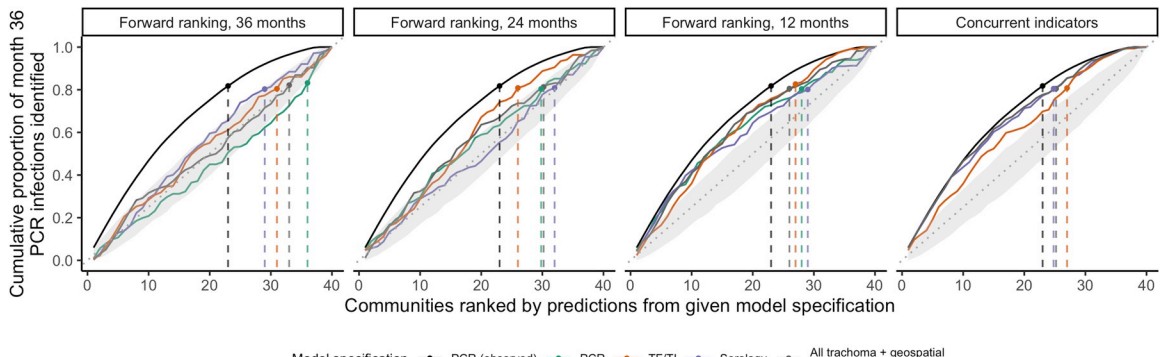

**Fig 5. Cumulative proportion of *C. trachomatis* infections at month 36 identified by concurrent and forward prediction models.** Dashed lines indicate the point at which the cumulative proportion of identified *Ct* infections at month 36, scaled to represent a sample of 30 individuals per community, surpassed 80%. The black line in each facet represents the optimal ordering of scaled PCR infections at month 36. To simulate a null distribution, we estimated the cumulative proportion of infections identified for 1000 random orderings of the 40 communities and plotted the 95% pointwise envelope (gray shading). For concurrent and 24-month-forward predictions, models using serology only and PCR only, respectively, performed equally well to a model using all trachoma indicators, geospatial features, a Matérn covariance, and ensemble machine learning; vertical lines were offset slightly for visibility.

address variability in sample size, the number of Ct infections in each community was scaled to represent a sample of 30 individuals. At month 36, 80% of Ct infections were concentrated in just over half of the communities (23/40), and ordering communities by cross-validated concurrent predictions using seroprevalence identified infections more efficiently (i.e. in fewer communities, 25/40) than ordering them by predictions using active trachoma (27/40) (Fig 5). Performance declined when using predictors measured 12 months in the past, and communities ranked by most predictors measured 24 and 36 months in the past could not identify high-burden communities based on PCR infections at month 36 better than chance. The distinction between models was greater at month 0 when 80% of Ct infections were concentrated in just the top 15 of 40 (38%) of communities (S8 Fig).

## Discussion

We conducted a comprehensive study of repeated cross-sectional measurements of active trachoma, PCR-positive ocular *Ct* infections, and serological responses to *Ct* antigens over three years in 40 communities in the hyperendemic Amhara region of Ethiopia. In the absence of MDA during the study, ocular *Ct* infections surged and became increasingly dispersed across study communities. Based on empirical variograms and Moran's I, we observed weak evidence for global spatial clustering in trachoma indicators over the study region. Seroprevalence among children 0–5 years old aligned closely with PCR prevalence measured at the same time, highlighting the potential for serosurveillance as a monitoring tool that corresponds well with levels of ocular infection and is potentially easier to measure [56]. Predictive performance of all models declined with increasing temporal lag between outcome and predictor measurements. In this setting, remotely sensed demographic, socioeconomic, and environmental geospatial layers, a spatial Gaussian process with Matérn covariance, and stacked ensemble machine learning did not meaningfully improve predictive performance compared to models using only trachoma indicators. We also illustrate a potential application of predictive models to rank-order and therefore efficiently identify communities with high infection burden; we expect that this approach may be most useful when infections are concentrated in a small number of communities.

Identifying potential future trachoma hotspots is notoriously challenging and sometimes termed "chasing ghosts" by trachoma programs [17]. Our results underscore the difficulty of predicting community-level *Ct* infection prevalence even a year into the future, at least in the context of increasing transmission in the absence of MDA. Furthermore, our "forward prediction" models were trained on infection outcomes from the desired prediction time point and thus were potentially more optimistic than true "forecasting" models trained solely on historical data. Prior efforts to forecast district-level TF [18] and village-level PCR prevalence [17] have explored mechanistic and statistical models and observed modest performance, with one investigation concluding that models with the highest uncertainty resulted in the best predictive performance [17]. It remains unclear why future prediction of trachoma presents such a difficult challenge, though likely contributing factors include the stochasticity of rare events especially in near-elimination settings [57], biological unknowns in the complex natural history of trachoma [58], and the extended duration between survey measurements (often 6 months or greater). Models for other neglected tropical diseases have achieved some success in future prediction at the sub-district level, though often capitalizing on larger datasets. For example, a recent study developed models with over 80% accuracy for prediction of *Schistosoma mansoni* persistent hotspots (defined as failure of a village to reduce infection prevalence and/or intensity by specific thresholds) up to two years in the future in the context of decreasing prevalence [59]. In a setting with fairly stable transmission, a sub-district-level study for

visceral leishmaniasis reported 85.7% coverage of four-month-ahead 25–75% prediction intervals for case counts [60].

Our investigation builds upon an existing body of work characterizing the dynamics between clinical, serological, and molecular trachoma indicators. Reports at the district, village, and individual level have established that relatively high levels of active trachoma or ocular infections tend to correspond to higher seroprevalence and/or seroconversion rates [14,61–64]; post-elimination settings have been of particular interest, with populations often displaying little to no antibody response [15,26,65–69]. Our findings align with earlier studies reporting that active trachoma was more strongly correlated with infection prevalence in populations with ongoing transmission compared to populations in which transmission has been suppressed by MDA [70–72]; also in agreement with prior findings, we observed that TI was slightly, but not significantly, more closely correlated with infection prevalence compared to TF immediately following MDA (**S9 Fig**) [73].

We additionally found that seroprevalence among children 0–9 years old was more closely aligned with concurrent infection prevalence than active trachoma both immediately after and following several years without MDA. Moreover, we found that seroprevalence was more strongly correlated with PCR prevalence among children 0–5 years old compared to children 6–9 years old, especially in the context of recent MDA at month 0. The lower correlation among 6–9-year-olds is likely due to the discordance between dampened transmission due to MDA and high seroprevalence from past exposures in this older age group. In contrast, seroprevalence among younger children reflects more recent transmission patterns, an observation which has been leveraged to investigate potential recrudescence in Tanzania [74]. Our findings support a focus on children 0–5 years old as a key sentinel population for trachoma serosurveillance.

Interestingly, active trachoma maintained a fairly consistent and strong correlation (~0.7) with PCR prevalence at both 0 and 12 months into the future (**Fig 3C**) and was a slightly better predictor of PCR prevalence 12 and 24 months ahead compared to serology (**Fig 4**); however, these findings should be interpreted with caution due to substantial uncertainty in predictive performance across models and inconsistent trends in active trachoma (**Table 1**).

In general, we did not observe strong evidence of global spatial autocorrelation for trachoma indicators over the study region, though spatial structure in PCR prevalence appeared to increase slightly over the study period. A prior analysis over the entire Amhara region reported evidence of spatial autocorrelation in TF between villages within 25km bands [10], and another study of TF and TI in Southern Sudan detected residual spatial structure between villages at approximately 8 km, after adjusting for age, sex, rainfall, and land cover [75]. A larger number of existing studies have characterized spatial autocorrelation at a fairly small scale. Studies using household-level information identified spatial clustering at less than 2 km for bacterial load [6,9], ocular infection [8,9], and active trachoma [76]. Our ability to detect spatial structure may have been limited by the geographic distribution of the communities, which was determined by the main trial objectives rather than optimized for estimation of spatial model parameters, which often requires points fairly close to one another [77]. In our study, only 26 (out of 780) pairs of study communities were within 5 km of one another leading to wide uncertainty at small ranges and hindering our ability to assess fine-scale spatial clustering.

In addition to rainfall and land cover, studies have reported associations between active trachoma and distance to water source [10,78–80], temperature [7,79,81], altitude [79,81–84], markers of socioeconomic status [7,10,78,80,84,85], and markers of personal or household hygiene, such as facial cleanliness [7,10,78,80,85–92]. Fewer studies have examined *Ct* infections identified by PCR, but associations reported were generally similar [85,92,93]. Using

LASSO to down-select geospatial features, we included night light radiance (often a proxy for socioeconomic activity [94]) and precipitation in prediction models. However, these features were unable to predict infection prevalence better than an intercept-only model. Predictive power of geospatial variables may have been limited by relative homogeneity across the study area, and the relatively small number of communities likely limited the predictive performance of all models.

As with all secondary analyses, our data were constrained by the objectives and design of the original trial. For instance, communities were purposely selected to be rural, but not too remote, and close to a potential water point–as a result, our findings may not be generalizable to urban or very remote areas, and spatial interpolation across the study site should be interpreted cautiously. Furthermore, this study was conducted in a hyperendemic region with increasing trachoma transmission in the absence of MDA and may not generalize to lower transmission settings. Ethiopia's Amhara region presents a particularly stubborn elimination challenge, as seven consecutive years of MDA were unable to sustain control before the start of this study. It is unclear whether prediction would be more or less challenging in the context of low transmission; we may expect more predictability in a "steady state" environment, but stochasticity is also a defining characteristic of near-elimination disease dynamics [57]. As an additional sensitivity analysis, we included survey month as a covariate to assess potential benefits of repeated sampling in the context of changing transmission and found only a modest improvement in predictive performance (**S10 Fig**).

The methods used here may be extended to other surveys in which trachoma prevalence can be estimated at the cluster level, including those supported by the Global Trachoma Mapping Project and Tropical Data service which have typically relied on multi-stage sampling strategies with compact segment sampling within villages [95,96]. Additional steps towards programmatic implementation of serosurveillance for trachoma should focus on further development of survey design and analytic methods including model-based geostatistics, which has recently been applied to trachomatous trichiasis [97], cost-effectiveness analyses to weigh the benefits of targeted interventions against the costs of fine-scale monitoring, and consideration of integrated serosurveillance programs to enable scalability.

## Conclusions

Serological markers among children 0–5 years old may be well-suited for community-level trachoma monitoring given their objectivity, durability, relative ease of collection, and strong correlation with ocular *Ct* infection prevalence. While seroprevalence and active trachoma were both correlated with infection prevalence in the midst of high transmission in the absence of MDA, only seroprevalence was strongly associated with community-level infections in the context of suppressed transmission directly following MDA. Accurate, future prediction of community-level *Ct* infection prevalence in settings with unstable transmission remains an open challenge.

## Supporting information

**S1 Table. Description and sources of geospatial variables explored for prediction analysis.**
(DOCX)

**S2 Table. Number of children evaluated across 40 study communities by trachoma indicator, age group and study month.**
(DOCX)

**S3 Table. Community-level seroprevalence across 40 study communities by antigen, age group, and study month.**
(DOCX)

**S1 Fig. Maps (A), variograms (B), and Moran's I (C) for seroprevalence among 0–5-year-olds at each study month.** Maps display prevalence for 40 study communities at each follow-up visit, spatially interpolated over the convex hull using kriging. Variograms capture similarity between community-level prevalence measurements as a function of distance between community pairs (in km), with smaller semivariance values representing increased similarity. Exponential (magenta) and Matérn (green) models were fit to each empirical variogram, and the effective range (dashed vertical line) is defined as the distance at which the fitted model reaches 95% of the sill. The Monte Carlo envelope (gray shading) displays pointwise 95% coverage of 1000 permutations, representing a null distribution. Moran's I was calculated over 1000 permutations (gray bars, with observed value represented by red line), and a permutation-based p-value was calculated. The base map layer for panel A in this figure was downloaded from Stamen Maps ("Terrain") and is available under the CC BY 3.0 license.
(TIF)

**S2 Fig. Maps (A), variograms (B), and Moran's I (C) for active trachoma prevalence among 0–5-year-olds at each study month.** Maps display prevalence for 40 study communities at each follow-up visit, spatially interpolated over the convex hull using kriging. Variograms capture similarity between community-level prevalence measurements as a function of distance between community pairs (in km), with smaller semivariance values representing increased similarity. Exponential (magenta) and Matérn (green) models were fit to each empirical variogram, and the effective range (dashed vertical line) is defined as the distance at which the fitted model reaches 95% of the sill. The Monte Carlo envelope (gray shading) displays pointwise 95% coverage of 1000 permutations, representing a null distribution. Moran's I was calculated over 1000 permutations (gray bars, with observed value represented by red line), and a permutation-based p-value was calculated. The base map layer for panel A in this figure was downloaded from Stamen Maps ("Terrain") and is available under the CC BY 3.0 license.
(TIF)

**S3 Fig. Correlations between PCR prevalence and antigen-specific seroprevalence by age group and over time.** Panels display Spearman rank correlations between community-level Pgp3 seroprevalence and PCR prevalence at months 0 and 36 (A), CT694 seroprevalence and PCR prevalence at months 0 and 36 (B), and PCR prevalence at month 36 and seroprevalence measured at each follow-up visit across 40 study communities (C). Correlations are shown separately for 0–5-year-olds (green) and 6–9-year-olds (purple) when possible, and 95% confidence intervals were estimated from 1000 bootstrap samples. Serology data was not collected for a random sample of 6–9-year-olds at months 12 and 24.
(TIF)

**S4 Fig. Spatio-temporal distribution of LASSO-selected geospatial predictor variables.**
Variables were estimated for 240 grid cells of 2.5 x 2.5 arc minutes (approximately 20 km$^2$ at the median latitude of the study area). Daily precipitation (A) and monthly night light radiance (B) averaged over the year were included in the final set of prediction models. The base map layer for this figure was downloaded from Stamen Maps ("Terrain") and is available under the CC BY 3.0 license.
(TIF)

**S5 Fig. Cross-validated $R^2$ for models predicting community-level PCR prevalence among 0–5-year-olds at month 0 (A), at month 12 (B), at month 24 (C), at month 36 (D), and pooled across all months (E).** Cross-validated $R^2$ (coefficient of determination), 95% influence-function-based confidence interval, and cross-validated root-mean-square error (RMSE, text label) are shown for each model specification. Blocks of size 15x15km were used for 10-fold spatial cross-validation. (D) is equivalent to **Fig 4** in the main text and is included here for comparison.
(TIF)

**S6 Fig. Cross-validated $R^2$ for stacked ensemble models predicting community-level PCR prevalence at month 36 among 0–5-year-olds using various superlearner models.** Cross-validated $R^2$ (coefficient of determination), 95% influence-function-based confidence interval, and cross-validated root-mean-square error (RMSE, text label) are shown for each model specification. Blocks of size 15x15km were used for 10-fold spatial cross-validation.
(TIF)

**S7 Fig. Cross-validated $R^2$ for models predicting community-level PCR prevalence among 0–5-year-olds at month 36 using random 10-fold cross-validation (A), 10-fold spatial cross validation with 5x5 km blocks (B), 15x15 km blocks (C), and 20x20 km blocks (D), and leave-one-out cross-validation (E).** Cross-validated $R^2$ (coefficient of determination), 95% influence-function-based confidence interval, and cross-validated root-mean-square error (RMSE, text label) are shown for each model specification. (C) is equivalent to **Fig 4** in the main text and is included here for comparison.
(TIF)

**S8 Fig. Cumulative proportion of *C. trachomatis* infections at months 0 and 36 identified by concurrent prediction models.** The black line in each facet represents the optimal ordering of scaled PCR infections at each respective month. Dashed lines indicate the point at which the cumulative proportion of infections, scaled to represent a sample of 30 individuals per community, surpassed 80%. To simulate a null distribution, we estimated the cumulative proportion of infections identified for 1000 random orderings of the 40 communities and plotted the 95% pointwise envelope (gray shading). At month 36, a model using only serology performed equally well to a model using all trachoma indicators, geospatial features, a Matérn covariance, and ensemble machine learning; vertical lines were offset slightly for visibility.
(TIF)

**S9 Fig. Correlations between PCR prevalence and active trachoma by age group and over time.** Panels display Spearman rank correlations between community-level TF prevalence and PCR prevalence at months 0 and 36 (A), TI prevalence and PCR prevalence at months 0 and 36 (B), and PCR prevalence at month 36 and active trachoma measured at each follow-up visit across 40 study communities (C). TF prevalence included any child diagnosed with TF, regardless of TI status, and vice versa. Correlations are shown separately for 0–5-year-olds (green) and 6–9-year-olds (purple), and 95% confidence intervals were estimated from 1000 bootstrap samples.
(TIF)

**S10 Fig. Cross-validated $R^2$ for models predicting pooled community-level PCR prevalence among 0–5-year-olds at month 36 with survey month (time) modeled as a linear covariate or Gaussian process.** Cross-validated $R^2$ (coefficient of determination), 95% influence-function-based confidence interval, and cross-validated root-mean-square error (RMSE, text label) are shown for each model specification. Blocks of size 15x15km were used for 10-fold spatial

cross-validation. For predictions 36 months ahead, time could not be explicitly modeled as a linear covariate as all outcomes were measured at month 36.
(TIF)

## Acknowledgments

We would like to thank the WUHA study participants and field team without whom this research would not be possible. We would also like to thank Abbott for its donation of the m2000 RealTime molecular diagnostics system and consumables. ***The findings and conclusions in this article are those of the authors and do not necessarily represent the official position of the Centers for Disease Control and Prevention. Use of trade names is for identification only and does not imply endorsement by the Public Health Service or by the U. S. Department of Health and Human Services.***

## Software

All analysis was conducted in R Version 4.0.2 ("Taking Off Again") [98]. The main R packages used for this analysis were *automap* (variograms) [99], *rgee* (Google Earth Engine) [100], *glmnet* (feature selection) [101], *spaMM* (regression with spatial Gaussian process) [102], *sl3* (stacked ensemble) [103], and *blockCV* (spatial cross-validation) [104].

## Author Contributions

**Conceptualization:** Christine Tedijanto, Benjamin F. Arnold.

**Data curation:** Christine Tedijanto, Jeremy D. Keenan.

**Formal analysis:** Christine Tedijanto.

**Funding acquisition:** Jeremy D. Keenan, Benjamin F. Arnold.

**Investigation:** Christine Tedijanto, Benjamin F. Arnold.

**Methodology:** Christine Tedijanto, Hugh J. W. Sturrock, Benjamin F. Arnold.

**Project administration:** Christine Tedijanto, Dionna M. Wittberg, Jeremy D. Keenan, Benjamin F. Arnold.

**Resources:** Solomon Aragie, Zerihun Tadesse, Mahteme Haile, Taye Zeru, Scott D. Nash, Sarah Gwyn, Diana L. Martin.

**Software:** Christine Tedijanto.

**Supervision:** Thomas M. Lietman, Jeremy D. Keenan, Benjamin F. Arnold.

**Validation:** Christine Tedijanto.

**Visualization:** Christine Tedijanto.

**Writing – original draft:** Christine Tedijanto, Benjamin F. Arnold.

**Writing – review & editing:** Christine Tedijanto, Solomon Aragie, Zerihun Tadesse, Mahteme Haile, Taye Zeru, Scott D. Nash, Dionna M. Wittberg, Sarah Gwyn, Diana L. Martin, Hugh J. W. Sturrock, Thomas M. Lietman, Jeremy D. Keenan, Benjamin F. Arnold.

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
