## [Decision Letter · Decision Letter 0]

29 Nov 2021

Dear Tedijanto,

Thank you very much for submitting your manuscript "Predicting future ocular Chlamydia trachomatis infection prevalence using serological, clinical, molecular, and geospatial data" for consideration at PLOS Neglected Tropical Diseases. As with all papers reviewed by the journal, your manuscript was reviewed by members of the editorial board and by several independent reviewers. In light of the reviews (below this email), we would like to invite the resubmission of a significantly-revised version that takes into account the reviewers' comments. 

We cannot make any decision about publication until we have seen the revised manuscript and your response to the reviewers' comments. Your revised manuscript is also likely to be sent to reviewers for further evaluation.

Sincerely,

Ali M. Somily

Associate Editor

Dileepa Ediriweera

Deputy Editor

Reviewer's Responses to Questions

**Key Review Criteria Required for Acceptance?**

**Methods**

-Are the objectives of the study clearly articulated with a clear testable hypothesis stated?

-Is the study design appropriate to address the stated objectives?

-Is the population clearly described and appropriate for the hypothesis being tested?

-Is the sample size sufficient to ensure adequate power to address the hypothesis being tested?

-Were correct statistical analysis used to support conclusions?

-Are there concerns about ethical or regulatory requirements being met?

Reviewer #1: Yes to all, though I do not profess deep expertise in geostatistical modelling

Reviewer #2: Ok

Reviewer #3: Methods: in general, the description of the methods section needs more detail to determine what was done. 

Line 108: only 40 communities in a zone were selected. How was the selection done? How many total communities exist in Wag Hemra zone? Please elaborate here. While spatial analyses can provide interpolation, assumptions if the data are too sparce become more problematic. 

Line 111: Please provide the dates of the trial here so that it can be placed in context with MDA. 

Line 122:Please clarify for here and Line 257: Were 30 participants TOTAL from each community chosen, or 30 individuals in each age group, for a total of roughly 90 per community. 

Figure One. A map of all the communities, with the ones enrolled in the study in red, should be created. A visual picture of the potential for clustering would be ideal. This would also help us understand the paucity of villages selected in the south and southeastern areas of the study area if in general that area , which appears mountainous, is sparsely settled. 

Line 125: Does molecular refer to a test for infection? Molecular tests are relatively nonspecific (for example, they are used for antibody testing as well) so perhaps using a term with more specificity would be useful or define its use here. 

Line 126: it is always tricky attempting to use clinical trial data for other purposes, and especially to combine data across arms. The authors state there was no difference in endpoint of community level ocular Ct infection among 0-5 year-olds between intervention arms-but is this based on a reduced sample size (since the age group six years and older is not included)? Are the point estimates very close as well? And what about the 6-9 year-olds? I would suggest they adjust for intervention arm as a variable, as I am not convinced yet that combining the data is across arms is justified. 

Line 157: This reviewer is concerned that inferences on trachoma , infection, and serologic status are being made per community on the basis of 30 children, when it takes almost 2500 to have confidence in an estimate made in a district. For these correlations, how is the error of measurement in the underlying estimate of interest propagated into the correlation itself? And if not, why not? 

Line 135: Please provide more information on the training of the graders. How many graders were involved, what was the training they received? Importantly, were the graders assigned to communities at random, or were they clustered in the woredas? If so, what was the agreement between them in grading? 

Line 140: Were air swabs taken to control for contamination in the field/lab? If so, what were the findings? Please provide the criteria by which pools were selected for unpooling, including the “other characteristics” and how often this occurred. 

Line 159: Apparently predictive correlations were done based just on the 0-5 year-olds, and this should be stated here. 

Somewhere in the methods, please provide a sample size estimate or at least a power calculation given the number of children per community and 40 communities.

**Results**

-Does the analysis presented match the analysis plan?

-Are the results clearly and completely presented?

-Are the figures (Tables, Images) of sufficient quality for clarity?

Reviewer #1: Yes to all

Reviewer #2: Ok

Reviewer #3: Results: 

Line 257. Please provide test of significance for the statement that seropositivity increased among 0-5 year-olds, because unless the test for trend is significant, I do not see a significant difference from baseline to 36 months.

Figure 2b: shouldn’t the title be Semivariogram, if the figure is the spatial dependence of the semivariance as stated? 

Figure 3 and Figure 4. Perhaps I am reading this incorrectly, but it appears to me that clinical TF/TI at 24 months correlates better with PCR outcomes at 36 months than does serology or even PCR at 24 months. This is worth mentioning, because the readers will notice it and want to have the authors take on the finding. Although serology was more correlated in the concurrent measurements, if one is trying to determine who is at risk going forward, could clinical TF one year earlier provide clues? This is also worth mentioning in the discussion, and the authors thoughts on if the correlation is strong enough, correlation of 0.75, to be useful. 

Paragraph starts line 349 : I am concerned that in this setting of very high rates of infection, while high burden communities are truly in need of MDA, it does not mean the other 17 communities do not have worrisome infection. Please show the infection rates in the other 17 communities. By using this approach to identify high burden, what is the likelihood that we are missing communities for intervention that still have infection rates greater than 5%, for example? Would another approach that identifies the communities that have very low infection (efficient identification of low burden communities) be a better way to screen out communities?

**Conclusions**

-Are the conclusions supported by the data presented?

-Are the limitations of analysis clearly described?

-Do the authors discuss how these data can be helpful to advance our understanding of the topic under study?

-Is public health relevance addressed?

Reviewer #1: Yes to all

Reviewer #2: Ok

Reviewer #3: limitations section is needed, the public health relevance needs more discussion as outlined below

Line 466. Conclusions: this reviewer is concerned with a conclusion that serological markers may be well suited for community level monitoring, without a fuller discussion on the practical implementation of the suggestions. At least a discussion of the need to study cost and resource balance of undertaking monitoring at community level versus precision targeting of antibiotic is needed. At present, districts will need a sampling approach to monitor some communities at some time points, and at each time point always in a position of not including some communities, which has an unknown effect on outcomes. I recognize that this study was not a true district study and was not designed to test how to implement seroprevalence strategically in a district, but in fact given the authors conclusions, some attention to this issue is warranted.

**Editorial and Data Presentation Modifications?**

Reviewer #1: (No Response)

Reviewer #2: (No Response)

Reviewer #3: (No Response)

**Summary and General Comments**

Reviewer #1: Trachoma is an important public health problem. This is an important addition to the literature on it. It’s an elegant study and has been very well written-up. To provide some sense that I have done my job in providing peer review, I am forced to provide a list, below, of what amounts to extremely minor comments. But also: I am a pedant. 

Great paper, though. I thoroughly enjoyed reading it.

The authors are quite right to use as their outcome variable the prevalence of conjunctival C. trachomatis infection rather than, say, the prevalence of TF. Although programmes currently use the prevalence of TF in 1-9-year-olds as a measure of success, this reflects the fact that when targets for trachoma programmes were being defined, assays for C. trachomatis infection were felt to be either too expensive, too insensitive or too unavailable to be used to monitor progress. Decisions on whether or not to apply antibiotic MDA should absolutely be based on the presence or absence of significant levels of transmission of the organism targeted by the antibiotic in question.

Line 81: please change “prevalence above 5%” to prevalence ≥5%”

Line 122: please change “three age groups (0-5, 6-9, 10+)” to “three age groups (0-5 years, 6-9 years, 10+ years)”

Line 122: I think the authors mean “thirty individuals in each of three age groups…” Is that correct? If yes, suggest amend the sentence.

Line 124: suggest delete “old” since the word “aged” already appears before the numbers

Line 135: please change the “X” to a multiplication sign

Lines 136 and 137: on each line, please change the hyphens after “inflammation” to em-dashes, without flanking spaces, as recommended in the report of the 4th Global Scientific Meeting on Trachoma

Line 137-138: please change “An individual was considered positive for clinical trachoma if either TF or TI was detected” to “An individual was considered positive for active trachoma if either TF or TI was detected”

Line 216: suggest change one of the “also”s in this sentence to another word. For example, “We additionally explored…”

Lines 253-255: The abbreviations TF and TI have already been defined, and “clinical disease” (by which the authors mean [active trachoma]: please see note above) has, too. The definitions of the signs TF and TI have not yet been given. It might be better to add this material to the section that is currently at lines 136-137. Please note that to be significant, follicles contributing to a diagnosis of TF have to (a) be in the central part of the upper tarsal conjunctival (not just “on the upper eyelid”) and (b) be at least 0.5mm in diameter; to be significant, the inflammatory thickening of TI has to be “pronounced” such that more than half of the normal deep tarsal vessels are not visible because they are obscured by inflammatory infiltration. I think it is worth adding these details

Lines 255, 270, 297-298, 302, 308, 324-325, 375-376, 413, 417, 422, 436, 443-444, abstract line 36: please change “clinical disease” or “clinical trachoma” to “active trachoma”

Table 1: the word “clinical” in front of “TF/TI” in the column heads is redundant; please delete

Line 269: please change “active ocular infection” to just “ocular infection”, or even better, “conjunctival infection” (since only the conjunctiva was sampled). Adding the adjective “active” risks confusion between [active trachoma] and [infection], which is already too widespread. The word “active” should also be deleted from lines 378 and 383.

Line 296: can the authors please provide a little more numerical clarity to the expression “very likely”?

Line 312: suggest change “serology data was” to “serology data were”

Anthony Solomon

Reviewer #2: Well written and good focused study on Predicting future ocular Chlamydia trachomatis infection prevalence using serological,

clinical, molecular, and geospatial data.

Reviewer #3: Title. The title should make clear that this is prediction at community level. Trachoma programs focus on district level surveillance, using sampling to represent the district. Identifying communities within districts is another level of complexity altogether. 

Abstract: 

Line 31: Please indicate if the communities were originally chosen at random or not. If not, the ramifications of this need to be addressed in the discussion. 

Line 33. The baseline findings need to be placed in context by indicating that MDA occurred 6 months prior to baseline. Infection would be much lower due to MDA, but disease burden would not yet have declined, as it does by month 12 in both age groups. I would suggest not devoting too much abstract space to baseline findings but rather focus on 24-month associations and then the concurrent associations. 

Line 42: the conclusion that serologic markers may be a programmatic tool has not been shown by this study, unless the authors discuss or theorize how this might be implemented in a programmatic setting. The evaluation of this tool in this study was done in a sample of communities, but how it might be extended to manage detection in communities not sampled is unclear. 

Introduction

Line 76: The authors nicely describe the evidence of trachoma clusters at village level, and high burden villages that are not apparent in overall EU estimates have been shown in previous studies. However, to argue that fine scale estimates of trachoma at community level could target limited resources is to ignore the fact that significant resources would be needed to undertake these estimates before targeting allocation of resources to communities with the highest burden. This issue needs to be addressed in the discussion, as a nuance to the rationale as stated here. 

Discussion 

The discussion section is balanced and well written. It would benefit from more discussion as suggested in the comments above. In addition, a discussion would be helpful of why the 6-9 year-olds may not have been as informative in a high prevalence setting ( they already had high seropositivity to begin with, and the increase in infection was more modest in this age range). Arguments bolstering the use of younger ages would be enhanced by citing a paper by Odonkor et al (Plos NTD 2021 ) where re-emergence was clearly evident from strategic serostatus determination in the youngest children, who should have had no exposure to infection if the program has lowered transmission.

A section on limitations would also be reasonable to add, particularly the issues with small numbers of communities.

PLOS authors have the option to publish the peer review history of their article (what does this mean?). If published, this will include your full peer review and any attached files.

Reviewer #1: No

Reviewer #2: Yes: Mehmet Sarier

Reviewer #3: No
---

## [Decision Letter · Decision Letter 1]

12 Feb 2022

Dear Tedijanto,

Thank you very much for submitting your manuscript "Predicting future community-level ocular Chlamydia trachomatis infection prevalence using serological, clinical, molecular, and geospatial data" for consideration at PLOS Neglected Tropical Diseases. As with all papers reviewed by the journal, your manuscript was reviewed by members of the editorial board and by several independent reviewers. The reviewers appreciated the attention to an important topic. Based on the reviews, we are likely to accept this manuscript for publication, providing that you modify the manuscript according to the review recommendations. 

Sincerely,

Ali M. Somily

Associate Editor

Dileepa Ediriweera

Deputy Editor

Reviewer's Responses to Questions

**Key Review Criteria Required for Acceptance?**

**Methods**

-Are the objectives of the study clearly articulated with a clear testable hypothesis stated?

-Is the study design appropriate to address the stated objectives?

-Is the population clearly described and appropriate for the hypothesis being tested?

-Is the sample size sufficient to ensure adequate power to address the hypothesis being tested?

-Were correct statistical analysis used to support conclusions?

-Are there concerns about ethical or regulatory requirements being met?

Reviewer #1: Yes, yes, yes, yes, yes; no concerns about ethical or regulatory requirements

Reviewer #2: none

Reviewer #3: thank you for the detailed response, the changes have improved the clarity greatly. Please insert in Line 110 (of marked up manuscript) that the "communities were not selected at random. They were....". This makes it clear these are not a random sample and the readers do not have to infer.

**Results**

-Does the analysis presented match the analysis plan?

-Are the results clearly and completely presented?

-Are the figures (Tables, Images) of sufficient quality for clarity?

Reviewer #1: Yes, yes, yes

Reviewer #2: none

Reviewer #3: Line 322-325 of marked up manuscript. The addition of the cohort study is new, and not described in the methods. Cohort studies are quite complex and previous work shows antibody seropositivity stability is highly correlated with age even in the under five year olds. Unless there is a description (other than in the title of the figure) of how these children were selected at random, an analysis of the age distribution of the seropositives at baseline, how many were followed longitudinally, etc, these data should not be included as they are difficult to interpret. They add little to the manuscript and should be deleted along with figure S4.

**Conclusions**

-Are the conclusions supported by the data presented?

-Are the limitations of analysis clearly described?

-Do the authors discuss how these data can be helpful to advance our understanding of the topic under study?

-Is public health relevance addressed?

Reviewer #1: Yes, yes, yes, yes

Reviewer #2: none

Reviewer #3: Very nice job with a full description of limitations

**Editorial and Data Presentation Modifications?**

Reviewer #1: The response to reviewer 3's comment about line 157 ("This reviewer is concerned that inferences on trachoma , infection, and serologic status are being made per community on the basis of 30 children, when it takes almost 2500 to have confidence in an estimate made in a district. For these correlations, how is the error of measurement in the underlying estimate of interest propagated into the correlation itself? And if not, why not?") could perhaps have taken into account the fact that surveys supported by GTMP/Tropical Data have, primarily for logistical reasons, almost always used compact segment sampling as the mechanism for selecting households and individuals within selected villages. That has the very helpful secondary advantage of meaning that cluster-level disease proportions are true cluster-level prevalences, assuming low levels of absenteeism, refusal and misdiagnosis. This point could perhaps be included in the discussion if the authors wished to do so, since it means that the methods that they propose could have even more applicability in the general case than in the specific dataset employed here (since the latter used a different sampling strategy at village level).

Reviewer #2: none

Reviewer #3: as noted above in two comments

**Summary and General Comments**

Reviewer #1: I am satisfied with the changes that the authors have made to the paper in response to the first round of review.

Reviewer #2: well wirtten good focused following revision.

Reviewer #3: Overall, excellent revision and response to reviewers. Addition of one sentence and deletion of non-essential few lines as noted above are final comments.

PLOS authors have the option to publish the peer review history of their article (what does this mean?). If published, this will include your full peer review and any attached files.

Reviewer #1: No

Reviewer #2: No

Reviewer #3: No

Figure Files:

Data Requirements:

Reproducibility:

References

---

## [Editor Report · Decision Letter 2]

23 Feb 2022

Dear Tedijanto,

We are pleased to inform you that your manuscript 'Predicting future community-level ocular Chlamydia trachomatis infection prevalence using serological, clinical, molecular, and geospatial data' has been provisionally accepted for publication in PLOS Neglected Tropical Diseases.

Best regards,

Ali M. Somily

Associate Editor

Dileepa Ediriweera

Deputy Editor

---

## [Editor Report · Acceptance letter]

8 Mar 2022

Dear Tedijanto,

We are delighted to inform you that your manuscript, "Predicting future community-level ocular *Chlamydia trachomatis* infection prevalence using serological, clinical, molecular, and geospatial data," has been formally accepted for publication in PLOS Neglected Tropical Diseases.

Best regards,

Shaden Kamhawi

co-Editor-in-Chief

Paul Brindley

co-Editor-in-Chief
